# Acetylation in Mitochondria Dynamics and Neurodegeneration

**DOI:** 10.3390/cells10113031

**Published:** 2021-11-05

**Authors:** Jaylyn Waddell, Aditi Banerjee, Tibor Kristian

**Affiliations:** 1Department of Pediatrics, University of Maryland School of Medicine, Baltimore, MD 21201, USA; JWaddell@som.umaryland.edu (J.W.); aditi.banerjee@som.umaryland.edu (A.B.); 2Veterans Affairs Maryland Health Center System, 10 North Greene Street, Baltimore, MD 21201, USA; 3Department of Anesthesiology and the Center for Shock, Trauma, and Anesthesiology Research (S.T.A.R.), University of Maryland School of Medicine, Baltimore, MD 21201, USA

**Keywords:** mitochondria, dynamics, tubulin, acetylation

## Abstract

Mitochondria are a unique intracellular organelle due to their evolutionary origin and multifunctional role in overall cellular physiology and pathophysiology. To meet the specific spatial metabolic demands within the cell, mitochondria are actively moving, dividing, or fusing. This process of mitochondrial dynamics is fine-tuned by a specific group of proteins and their complex post-translational modifications. In this review, we discuss the mitochondrial dynamics regulatory enzymes, their adaptor proteins, and the effect of acetylation on the activity of fusion and fission machinery as a ubiquitous response to metabolic stresses. Further, we discuss the role of intracellular cytoskeleton structures and their post-translational modifications in the modulation of mitochondrial fusion and fission. Finally, we review the role of mitochondrial dynamics dysregulation in the pathophysiology of acute brain injury and the treatment strategies based on modulation of NAD^+^-dependent deacetylation.

## 1. Introduction

A mitochondrion is an intracellular organelle that represents an evolutionary product of endosymbiosis, which occurred by fusion of an anaerobic bacterial cell and a proteobacterium possessing a respiratory apparatus with elements of the respiratory chain, the tricarboxylic acid cycle (TCA), and oxidative phosphorylation [1,2,3,4,5,6]. In this symbiosis, the mitochondrion provided the host cell with aerobically produced energy, while the host cell synthesized metabolites required for the mitochondrial respiration. Due to the endosymbiotic relation, the pro-eucaryotic cells had an energetic and therefore evolutional advantage to survive under the conditions of gradually increasing oxygen concentration in the atmosphere [1,2,3,4,5,6].

The uniqueness of mitochondria is not only in their evolutionary origin but also in their complex functional diversity. Although these organelles are mainly associated with cellular bioenergetic metabolism, they also play an essential role in the cellular metabolism of macromolecules including lipids, protein, DNA, and RNA. Furthermore, mitochondria play a significant role in the regulation of the cellular response to oxidative stress, and changes in metabolic demands ([7], for review see [8]). Finally, mitochondria have a critical role in regulating cellular calcium homeostasis [9,10,11,12,13,14]. In contrast to other organelles, mitochondria cannot be formed de novo and have to be generated from pre-existing organelles and mitochondrial growth [15]. To efficiently perform their function and successfully assist the cells in managing metabolic challenges associated with changing environment or stress conditions, mitochondria possess features of a dynamic organelle; thus, they can move and go through cycles of fission and fusion [16,17,18,19,20,21,22]. These features give the mitochondria the ability to deliver ATP directly where spatially within a cell energy is most required. This is particularly essential for brain cells with complex morphology and an abundant number of processes. Furthermore, the fission of mitochondria ensures a proper inheritance and distribution of mitochondria to sister cells during division, and by fragmenting the mitochondria into smaller organelles, the fragments containing damaged structures or malfunctioning proteins can be efficiently removed by mitophagy [23,24].

There are dedicated proteins that are responsible for the fission and fusion of mitochondria. The fission is orchestrated by membrane-associated adaptors and modulated by organellar and cytoskeletal interaction. The activity of proteins that control the mitochondrial dynamics is modulated by post-translational modifications. One of the ubiquitous modifications that is mainly affected by changes in cell bioenergetic metabolism is acetylation. In this review, we discuss the effect of acetylation of fusion/fission proteins, and cytoskeletal proteins involved in mitochondrial dynamics.

## 2. Mitochondrial Fission Controlling Machinery

Proteins controlling mitochondrial fission and fusion interact with the outer and inner mitochondrial membrane in a process that leads either to the division of the organelle into separate individual mitochondria or leads to combining of two organelles into one larger mitochondrion. The executioner of mitochondrial fission is the GTPase of the dynamin superfamily of proteins, known as dynamin-related protein (Drp1). Drp1 requires adaptor proteins to secure the interaction with the mitochondrial outer membrane. Mitochondrial fission 1 (Fis1), mitochondrial fission factor (Mff), and mitochondrial dynamics protein 49/51 (MiD49/51) were identified as adaptor proteins that interact with Drp1 [25,26]. Fis1 has also been implicated as a protein involved in mitophagy, a process that removes dysfunctional mitochondrial organelles [27,28]. However, mitophagy is preceded by mitochondrial fragmentation generating several smaller organelles that can be fused with autophagosomes. Mitochondria that are dysfunctional lose their membrane potential and that is followed by ubiquitination of outer membrane proteins by ubiquitin ligase Parkin or MARCH5 leading to initiation of mitophagy [29,30]. Mff is a tail-anchored protein that integrates into the mitochondrial outer membrane and also the peroxisomal membrane [31,32]. MiD49 and MiD51 have N-terminal transmembrane anchors for specific interaction with mitochondrial membrane [33]. Each adaptor can independently recruit Drp1 into the outer mitochondrial membrane and their loss results in dysfunctional mitochondrial fission [34,35,36,37]. By generating smaller organelles, this process enables mitochondria to move more efficiently within the cell and deliver ATP where it is most needed. Finally, during cell division, mitochondrial fission ensures an equal distribution of mitochondrial mass into daughter cells [24,38].

### Proteins Regulating the Fusion Process

Due to extensive fission, the mitochondrial organelles can reach a submicron size that significantly limits their ability to carry out the bioenergetic functions due to severely reduced availability of essential enzymes and metabolites [39]. To prevent the collapse of cellular bioenergetic metabolism due to over-fragmentation of mitochondrial organelles, there is an opposing fusion process that combines individual organelle content [40]. Thus, the fusion is required for remixing of small organelle content to ensure a sufficient level of enzymes, intermediate metabolites, and number of mitochondrial DNA copies for sustainable and efficient organelle functioning [41]. During the fusion process, two events occur. First, the outer membrane of the two neighboring mitochondria fuse, followed by fusion of their inner membranes. The outcome is a transfer of matrix content through the exchange of mtDNA, proteins, lipids, and metabolites between the newly fused organelles [25,42]. The fusion process is carried out and controlled by three GTPases. Mitofusins 1 and 2 (Mfs1, Mfs2) that are responsible for outer mitochondrial membrane (OMM) fusion, and optic atrophy 1 protein (OPA1), which regulates the conformational changes of the inner mitochondrial membrane (IMM) [43]. Mitofusins are directly involved in the docking and fusion of OMM. The docking activity relies on their ability to self-assemble into oligomers [44]. It has been shown that only deletion of both mitofusins isoforms completely inhibits OMM fusion [45]. However, Mfn1 knockout leads to increased mitochondrial fragmentation, while Mfn2 knockout leads to mitochondrial swelling, resulting in a spherical shape [45]. This suggests that Mfn1 and Mfn2 have different roles in the fusion process. Mfn2 is also involved in axonal transport of mitochondria via interaction with Miro protein, an adaptor required for mitochondrial transport along the neuronal axons [46]. Furthermore, both mitochondrial division and fusion occur at contact sites of endoplasmic reticulum (ER) and mitochondria, and it was suggested that Mfn2 mediates ER-mitochondria tethering [47,48]. The mitofusins also interact with adaptor proteins on the OMM to carry out the fusion process. Misato (MSTO1) has been reported to be involved in OMM fusion [49], and it was suggested that this adaptor protein can support mitochondrial fusion through initiation of the fusion process [50,51].

Inner mitochondrial membrane fusion is controlled by dynamin-like GTPase OPA1. Opa1 activity is directly regulated by proteolytic cleavage. Once this protein is imported into mitochondria, the mitochondrial processing peptidase cleaves the mitochondria targeting sequence, producing the long isoforms (L-OPA1) [52,53,54,55]. OPA1 controls both mitochondrial IMM fusion and cristae morphology [52,54,56,57]. L-OPA1 isoforms can be further proteolytically cleaved into short forms (S-OPA1) soluble in the intermembrane space. There are two proteolytic cleavage sites where the OPA1 protein is cleaved by mitochondrial metallo-endopeptidases OMA1, and ATP-dependent zinc metalloprotease YME1L [58,59,60]. Both S-OPA1 and L-OPA1 are present in mitochondria under normal, physiologic conditions; however, the pro-fusion activity of OPA1 is mostly associated with the L-OPA1 isoform. OMA1 is activated upon stress insults, resulting in degradation of L-OPA1 and leading to mitochondrial fragmentation [61,62,63]. The role of the two OPA1 versions in the fusion process is not clear. Initially, it was reported that both L-, and S-OPA1 are required for the fusion [64]. However, other reports suggested that only the L-OPA1 is the profusion form [52,59,65]. As mentioned above, the S-OPA1 is produced after proteolytic cleavage of L-OPA1 activated by stress or pathologic conditions. The increase in the soluble S-OPA1 levels was observed following global cerebral ischemia [39,66] and neonatal hypoxic-ischemic brain injury [67]. This was accompanied by increased fragmentation of mitochondria in brain cells. Similarly, reduced levels of L-OPA1 were associated with cessation of mitochondrial fusion and mitochondrial cristae dilation after excitatory insult [68]. Overexpressing OPA1 reversed the changes in mitochondrial morphology and increased neuronal survival after excitotoxic insult [68]. Generation of the short OPA1 isoform along with fragmentation of mitochondria is favored at low membrane potential and low ATP levels [52,64,69,70]. Thus, conversion of OPA1 to the short isoform inhibits the fusion machinery of dysfunctional organelles and prevents the fusion of defective organelles with the functioning mitochondrial network, contributing to quality control [60,61].

## 3. Tight Regulation of Fission and Fusion Proteins Activity by Post-Translational Modifications

The interaction of the adaptor proteins with Drp1 is fine-tuned by post-translational modifications including SUMOylation, ubiquitination, phosphorylation, S-nitrosylation, O-linked-N-acetyl-glucosamine glycosylation, and acetylation (for review see [24]). Phosphorylation was identified as the major post-translational modification that regulates Drp1. There are two serine residues that are phosphorylated in Drp1. Phosphorylation of Ser616 activates the Drp1 binding to the adaptor proteins in the outer mitochondrial membrane and leads to increased mitochondrial fission, whereas phosphorylation of Ser637 prevents the Drp1 activation [24,71]. Fission-related adaptors are also subject to post-translational modifications. The March5 ubiquitin ligase regulates the levels of MiD49, and MFF is phosphorylated via AMP-activated protein kinase (AMPK) under stress conditions [72,73]. Another ubiquitous post-translational modification that modulates the activity of fusion and fission protein is lysine acetylation.

### 3.1. Role of Acetylation in Control of Mitochondrial Dynamics

Acetylation of histone and non-histone proteins is a post-translational modification that has been known for decades [74,75,76]. During acetylation, the acetyl group donated by acetyl-CoA (AcCoA) is transferred to a lysine residue of the target protein by histone acetyltransferases (HATs). This process is highly reversible by activity of histone deacetylases (HDACs). HDACs can be broadly divided into four classes. Class I is represented by HDACs (1,2,3 and 8), in class II HDACs 4,5,6,7,9, and 10 is identified, class III HDACs are also called sirtuins (SIRT1-SIRT7), and finally, HDAC 11 represents class IV [77].

Sirtuins require NAD^+^ for their activity and modulate protein signaling and function by transferring the acetyl group from the lysine of acetylated protein to the ADP-ribose moiety of NAD^+^. During this process nicotinamide is cleaved from NAD^+^ and *O*-acetyl-ADP-ribose is generated [78]. SIRT1, SIRT6, and SIRT7 are localized in the nucleus [79], SIRT2 is considered to be a cytosolic protein [80], and SIRT3-5 were identified as mitochondrial proteins [81,82,83,84]. However, only SIRT3 plays a key role in deacetylation of mitochondrial proteins [81,82]. SIRT4 shows ADP-ribosyl transferase activity [83,85] and SIRT5 acts as desuccinylase [86,87] (see Table 1).

Histone acetyltransferases (HATs) are generally categorized as nucleic or cytoplasmic HATs [94,95]. Mitochondrial proteins are acetylated by mitochondria-specific acetyltransferase, general control amino acid synthesis 5-like 1 (GCN5L1) [96]. GCN5L1 is a non-transmembrane protein located within mitochondria [96].

Recently, it was reported that Drp1 can also be acetylated at lysine 642 (K642) [97]. The increased acetylation of Drp1 in heart mitochondria was observed under conditions of a high-fat diet that resulted in accumulation of AcCoA and increased total protein acetylation including Drp1. The Drp1 acetylation was coupled with phosphorylation of Drp1 at S616 and its translocation to mitochondria (Figure 1). High-fat diet also significantly decreased total intracellular NAD^+^ content. Restoration of NAD^+^ levels prevented the Drp1 hyperacetylation and blocked the Drp1 phosphorylation at S616, suggesting that NAD^+^—mediated acetylation regulates Drp1 protein fission activity. Thus, Drp1 acetylation at K642 was required for increased Drp1 phosphorylation at S616, which promoted its mitochondrial translocation and fission activity [98]. Although in this work the deacetylase that was responsible for removing the acetyl group Drp1 was not identified, the finding that this process was NAD^+^ dependent suggests an involvement of cytosolic NAD^+^-dependent deacetylase SIRT2.

The fission process is controlled not only by the Drp1 acetylation but also by the acetylation of adaptor protein Fis1. For example, decreased cellular AcCoA levels due to increased lipid generation by acetyl-CoA carboxylase alpha (ACC1) leads to enhanced mitochondrial fission [99]. This is because ACC1 activation reduces cellular AcCoA level, thereby inhibiting Fis1 protein acetylation, which blocks Fis1 polyubiquitination and subsequent proteasome degradation [99]. Thus, Fis1 effect on mitochondrial dynamics is fine-tuned by the interplay between acetylation and ubiquitination of these proteins.

Under pathologic conditions, Opa1 is also hyperacetylated. This reduces its GTPase activity, thus inhibiting the fusion process, increasing mitochondrial fragmentation [100]. Once Opa1 is deacetylated by SIRT3, the fusion activity of OPA1 is restored [100]. Similarly, acetylation status of Mfn1 also affects the fusion process. Under fasting conditions, Mfn1 is deacetylated by HDAC6 leading to mitochondrial hyperfusion [101] due to increased fusion activity of its deacetylated form. This process is linked to interaction of Mfn1 with MARCH5 causing subsequent Mfn1 ubiquitination. Thus, these data imply that the increased ubiquitination of Mfn1 is dependent on its acetylation status [102], and suggests a possibility that once the fusion process is carried out, Mfn1 is destined for degradation by the ubiquitin–proteasomal system (Figure 2).

### 3.2. Effects of Mitochondrial Protein Acetylation on Free Radical-Induced Fission and Fusion Dynamics

The fission activity of Drp1 is also stimulated by modifications induced via the interaction of the Drp1 protein with free radicals, particularly nitric oxide (NO). NO belongs to a family of reactive nitrogen species and signals primarily through the formation of S-nitrosothiols, representing S-nitrosylation [103]. Furthermore, NO reacts readily with superoxide generating highly reactive peroxynitrite, which can nitrate tyrosine residues to form nitrotyrosine [103]. The cysteine residue (Cys644) of Drp1 can be S-nitrosylated by nitric oxide (NO) leading to increased Drp1-induced fission [25].

One of the adverse side effects of mitochondrial respiration is the production of free radicals, mainly by the respiratory chain enzymes [104,105]. However, free radicals also serve as signaling molecules regulating several biochemical processes [106]. To maintain the ROS levels within physiological range mitochondria and cells possess enzymatic machinery that eliminates the excess of free radicals [104,107,108,109]. These antioxidant mechanisms in mitochondria include the matrix manganese superoxide dismutase (MnSOD), glutathione (GSH), glutathione reductase (GR), glutathione peroxidase (GPX), and the thioredoxin system [107,108,110,111]. MnSOD is essential in protection of mitochondria against oxidative stress since it detoxifies superoxide [110].

Acetylation affects the activity of the MnSOD [112,113,114]. Hyperacetylation of this enzyme under stress or pathologic conditions results in increased production of superoxide due to acetylation-induced inhibition of MnSOD [115,116,117]. The increased levels of superoxide facilitate peroxynitrite production and S-nitrosylation of DRP1, resulting in increased mitochondrial fragmentation [39,116]. The acetylation of MnSOD is controlled by NAD-dependent SIRT3 implying that changes in mitochondrial NAD^+^ pools can significantly impact superoxide levels and the mitochondrial fission process. Thus, the maintenance of physiological matrix NAD^+^ levels is essential not only for mitochondrial respiratory functions but also impacts mitochondrial dynamics [116]. This notion is supported by data showing that reduced mitochondrial NAD^+^ levels following ischemic insult lead to extensive mitochondrial fission that can be reversed by administration of NAD^+^ precursor nicotinamide mononucleotide (NMN) [116], and that SIRT3 knockout animals show extensively fragmented brain mitochondria when compared to wild type animals [105]. Thus, the NMN treatment prevents the pathologic post-ischemic NAD^+^ consumption, inhibits the increase in free radical production leading to recovery of mitochondrial shape and protects against ischemia-induced brain damage [116].

## 4. Role of Cellular Cytoskeleton in Mitochondrial Fission/Fusion Dynamics

To support the diverse functions of mitochondria within the cell, particularly in ones with complex morphology represented by neurons and glia, the mitochondria move along microtubules (MT) aided by motor proteins [118,119]. Actin microfilaments (MFs) also play a role in mitochondrial transport [120,121,122]. In cells, which lack MTs but retain essentially normal levels of MFs, mitochondria continue to move bidirectionally but the average mitochondrial velocity is reduced for both directions of movement [120]. However, in axons disassembling microtubules leads to complete elimination of mitochondrial motility [123]. In cells, which retain microtubules but are lacking microfilaments, the average mitochondrial velocity is increased [120]. Thus, mitochondrial transport can occur along both microtubules and microfilaments, but with different velocities.

Microtubules are cylindrical polymers composed of α- and β-tubulin heterodimers [124]. Their functional diversity can be regulated by post-translational modifications (PTMs) [125,126]. The structures of microtubules are stabilized by binding other proteins called microtubule-associated proteins (MAPs).

Interaction of mitochondria with microtubules and their movement depends on MAPs and their post-translational modifications [127]. MAPs can be divided into microtubule stabilizers and destabilizers, and molecular motors that are responsible for mitochondria movement [126,127]. Motor proteins include kinesins that drive anterograde transport and dyneins, which typically drive retrograde mitochondrial movement but also can be involved in bidirectional transport [118,128].

The major post-translational modifications that regulate microtubule cytoskeleton include detyrosination, Δ2-tubulin generation, glutamylation, glycylation, and acetylation [126]. Detyrosination denotes a reversible removal of C-terminal tyrosine and the detyrosinated tubulin can then be further converted into Δ2-tubulin [129]. This represents very stable MTs as a final stage of functional differentiation [130].

Glutamylation and glycylation are also a modification of the C-terminal tails of tubulins and provide a mechanism for fine-tuning the interactions between microtubules and their binding partners including motor proteins [131,132].

### 4.1. Actin Cytoskeleton Acetylation and Mitochondrial Fission

The fission of mitochondrial organelles is preferentially observed at the sites where there is a close contact of mitochondria with endoplasmic reticulum (ER), suggesting that the ER plays a significant role in mitochondrial division [133]. The interaction of ER and mitochondria during the fission process is mediated by actin polymerization and recruitment of Drp1. The actin polymerization is carried out through ER-bound actin assembly factor inverted formin 2 (INF2) [134] and mitochondrial Spire1C [135]. The polymerized actin promotes Drp1 recruitment to the mitochondrial surface. Actin polymerization is stimulated by mitochondrial calcium uptake from ER, associated with increased activity of mitochondrial respiratory chain [136] and is affected by actin-depolymerizing protein cofilin1 that acts as a negative regulator of mitochondrial Drp1 activity [137]. Actins contain several lysine residues that can be acetylated [138]. Complex of acetylated actin with cyclase-associated protein (CAP) inhibits INF2, and this inhibition can be released by HDAC6 [139]. Thus, actin acetylation affects its polymerization at ER–mitochondria contact sites and downstream can modulate the mitochondrial fission process.

### 4.2. Mitochondrial Movement and Tubulin Acetylation

Acetylation of α-tubulin at lysin 40 was reported as a specific post-translational modification that enhances recruitment of kinesin-1 and dynein/dynactin motor complexes to microtubules that results in increased anterograde and retrograde transport [140,141]. However, the mechanisms of this interaction are not clear since lysine 40 is localized on the inside of the microtubule polymer [142], whereas most interactions between microtubules and their associated proteins take place along the microtubules outer surface. It was suggested that the acetylation of the lysine 40 residue of α-tubulin modulates microtubule conformation by triggering subtle changes in the way tubulins interacts with each other [143,144] and in this way, stimulate the interaction with motor proteins [140,141,145].

Tubulin acetylation level is a result of a balance between the activity of the cytoplasmic deacetylases HDAC6 and SIRT2 and several acetyltransferases [146]. Alpha-tubulin acetyltransferase 1 αTAT1/MEC-17 was reported as a major acetyltransferase targeting tubulin [147,148,149]. It was shown that MEC-17 knockdown dramatically reduces tubulin acetylation under both basal or various stress-induced conditions, and it is required for stress-induced MT hyperacetylation [150]. HDAC6 and SIRT2 were reported to co-localize along the microtubule network and coimmunoprecipitate. Furthermore, tubulin hyperacetylation was observed after silencing of HDAC6 or SIRT2 alone suggesting that these deacetylases act interdependently in a protein complex [151]. In the brain, however, SIRT2 shows abundant expression, particularly during adult age [152], and HDAC6-deficient mice do not show increased tubulin acetylation in brain tissue when compared to wild-type animals [153]. However, another study did find increased α-tubulin acetylation in the brain of HDAC6 knockout animals [154]. Furthermore, there are conflicting results in the literature regarding the cell-type-specific distribution of HDAC6. High expression levels of HDAC6 were shown predominantly in Purkinje cells [155] (however, see [154]) and in neurons of dorsal and median raphe nuclei [156]. HDAC6 is expressed only in very few hippocampal and cortical neurons [156]. Nevertheless, HDAC6 inhibition promoted both retrograde and anterograde mitochondrial movement in hippocampal or motor neurons together with increased tubulin acetylation [157,158]. These conflicting results might reflect the age-dependent differences in HDAC6 expression levels in cell culture since the HDAC6 distribution in brain was examined using brain tissue sections from adult animals and the effect of HDAC6 inhibitors on mitochondrial movement was observed in primary neuronal tissue culture. Furthermore, downregulation of HDAC6 causes a reduction in the net activity of mitochondrial enzymes, including respiratory complex II and citrate synthase [159]. Acetylation of α-tubulin also modulates mitochondrial length by affecting the organelles transport activity and by modulating the interaction of mitofusin 2 with motor proteins [46]. Kim et al. [160] reported that motile mitochondria are longer than stationary ones, suggesting that the change in mitochondrial length is coupled with dynamics of mitochondrial transport. A list of mitochondrial dynamics control proteins that activity is modulated by acetylation is in Table 2.

These reports suggest that although mitochondrial dynamics are predominantly regulated by the fission and fusion proteins and their post-translational modifications, during the last step of the fission process the individual organelles need to physically move from each other, or to initiate fusion they need to gain close contact. This is ensured by the organelle’s movement. Thus, the transport of mitochondria along the cytoskeleton composed of microtubules and microfilaments represents an additional control step that is also fine-tuned by post-translational modifications of the cytoskeleton and motor proteins. However, as we mentioned above, the mitochondrial fission sites are found at ER and also at multiple organelles, including lysosomes and trans-Golgi network (TNG) vesicles [161,162]. Examination of the interaction between mitochondrial fission and the TGN vesicles suggests that these vesicles drive the last step of mitochondrial division [163]. Another possibility is that the smaller organelles can locally change their position driven by thermal energy, without the involvement of cellular transport machinery, to facilitate fission or fusion. However, this process is random, lacking any control or modulation by changes in intracellular metabolic conditions. Furthermore, data showing striking absence of any mitochondrial motility in cells lacking both microtubules and microfilaments suggest that the contribution of this type of movement to overall mitochondrial transport is probably negligible [120]. However, it might facilitate the final separation of vesicles during fission or help to initiate the outer mitochondrial membrane fusion.

## 5. Pathophysiology of Mitochondrial Dynamics Dysfunction

Since mitochondrial dynamics are integral to several essential processes in living cells, disturbance in the regulation of these mechanisms is associated with several neurodegenerative diseases or acute brain injury (for review see [19,146,165]). Excessive mitochondrial fragmentation can lead to a decline in mitochondrial bioenergetic activity, followed by cellular bioenergetic failure and ultimately cell death [19,24,39,66,166]. Increased, excessive fragmentation of mitochondria in neurons or astrocytes was reported following ischemic, traumatic or excitotoxic insult [39,66,167,168,169,170,171,172]. The notion that heavily fragmented mitochondria can lead to cellular pathophysiology and cell death is supported by the observation of differential, hippocampal subregion-specific changes in mitochondrial fission following ischemic insult [39,116]. After initial post-ischemic fragmentation, mitochondria in ischemia-resistant CA3 and dentate gyrus neurons can fuse and regain their pre-ischemic shape and length [39]. However, in the vulnerable CA1 neurons that die during reperfusion, mitochondrial fragmentation is not reversed, but progresses generating submicron size organelles [39]. Interestingly, administration of nicotinamide mononucleotide (NMN) following the ischemic insult leads to dramatic neuroprotection [173], and recovery of mitochondrial morphology in CA1 neurons [116].

### Role of NAD^+^ in the Regulation of Mitochondrial Dynamics in Acute Brain Injury

NMN is an endogenous metabolite that serves as a precursor for NAD^+^ synthesis by nicotinamide mononucleotide adenylyl transferase (Nmnat) enzyme [174]. Mitochondrial NAD^+^ levels in the brain are depleted following ischemic insult, and NMN treatment normalizes the post-ischemic NAD^+^ pools [116,173]. The changes in intracellular and mitochondrial NAD^+^ levels can affect the fusion and fission process via post-translational modifications (PTM) of the proteins that control the mitochondrial dynamics (see above). Thus, NAD^+^ can have a direct impact on the acetylation levels of mitochondrial enzymes via modulation of intramitochondrial deacetylase SIRT3 activity. Similarly, changes in cytosolic NAD^+^ levels can affect the Drp1 and microtubules acetylation via NAD^+^-dependent SIRT2 activity [89,97,155]. Indirectly, the intramitochondrial NAD^+^ affects the mitochondrial reactive oxygen species (ROS) production, and this way also impacts free radical-dependent post-translational modification of Drp1 [116,175]. The reduced mitochondrial NAD^+^ levels lead to oxidative stress and activation of mitochondrial fission [25,103]. Thus, ischemic insult depletes mitochondrial NAD^+^ pools and consequently increases mitochondrial protein acetylation that contributes to post-ischemic oxidative stress and shifts mitochondrial dynamics towards fission [116,175]. The role of mitochondrial SIRT3 activity was confirmed also by using SIRT3 knockout animals that showed hyperacetylation of mitochondrial proteins, higher free radical production, and more fragmented mitochondria when compared to wild type animals [116]. Furthermore, increased acetylation of Drp1 promotes its activity and causes mitochondrial fragmentation [97]. This can be reversed by supplementation of NAD^+^ precursor NMN [97]. These data thus suggest that also cytosolic NAD^+^ levels can alter mitochondrial morphology via acetylation of Drp1 most likely by modulation of SIRT2 activity. Similarly, since microtubules acetylation is one of the factors affecting mitochondrial movement and dynamics, the ischemia-induced reduction in microtubules stability due to decreased microtubules acetylation and reduced expression of MAPs is associated with post-ischemic cell death [176,177]. Acetylation effects on mitochondrial interaction with microtubules and related fusion/fission proteins are shown in Figure 3.

## 6. Conclusions

Acetylation is one of the most ubiquitous post-translational modifications that represents a universal mechanism of regulation of enzymes activity including one that controls the mitochondrial dynamics. The balance between fission and fusion is essential for mitochondria to work properly and maintain a healthy functioning cell.

Most studies examining the pathophysiology of mitochondrial dynamics focused on changes in expression levels of fusion and fission proteins or the aberrant changes in their posttranslational modifications. However, the physiological cycles of mitochondrial fission and fusion also depend on mechanisms that involve and control mitochondrial movement. Therefore, when examining treatment strategies, one should also consider targeting the mechanisms involved in mitochondrial movement and its regulation, including proteins associated with the interaction of mitochondria with other intracellular organelles. Thus, the therapeutical approach needs to target several mechanisms to be effective and have a significant impact.

## Figures and Tables

**Figure 1 cells-10-03031-f001:**
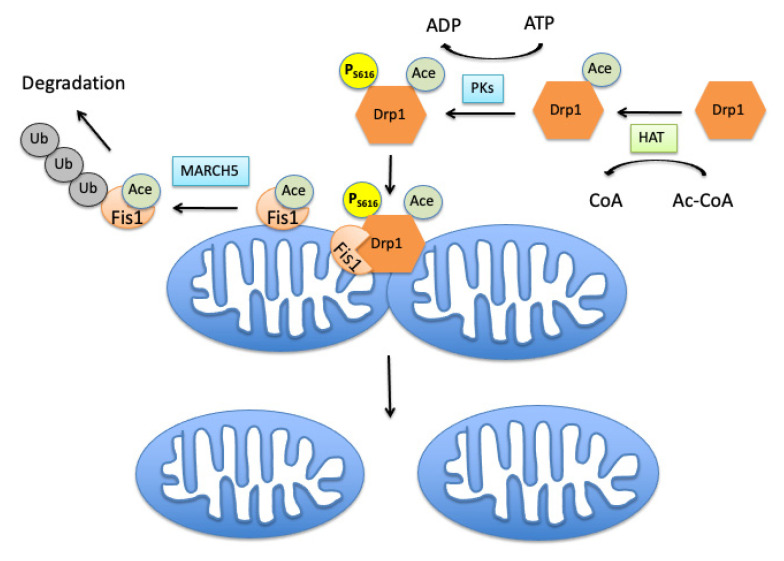
Acetylation of mitochondrial fission proteins. The acetylation of Drp1, a fission driving protein, by cytosolic histone acetyltransferase (HAT) enables its phosphorylation at serin 616 residue by protein kinases (PKs) and its translocation to mitochondria by binding to the Fis1 adaptor located in the outer mitochondrial membrane (OMM), thus promoting mitochondrial fission. Increased acetylation of Fis1 is linked to its ubiquitination by ubiquitin ligase MARCH5 and proteolytic degradation. HAT transfers the acetyl group from acetyl-CoA (Ac-CoA) to the target protein generating acetylated protein and coenzyme A (CoA).

**Figure 2 cells-10-03031-f002:**
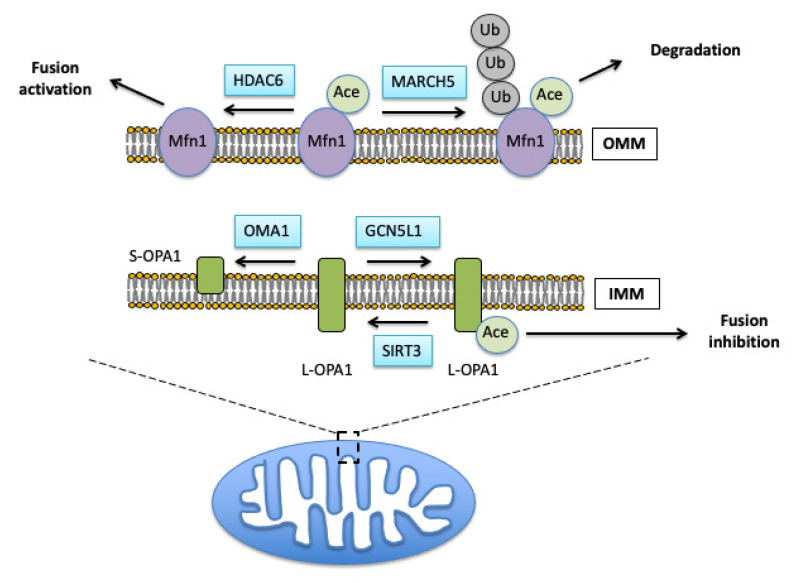
Acetylation of fusion regulating proteins. Acetylation of mitofusion1 (Mfn1) results in ubiquitination of this protein by MARCH5 and its proteasomal removal. Mfn1 is deacetylated by histone deacetylase 6 (HDAC6) leading to increased fusion activity. Hyperacetylation of the inner membrane (IMM) fusion protein OPA1 by mitochondrial acetyl transferase GCN5L1 results in inhibition of its fusion activity. Mitochondrial SIRT3 deacetylates OPA1 which has a profusion effect. OPA1 is cleaved by mitochondrial metallo-proteases OMA1 and YME1L into short isoforms.

**Figure 3 cells-10-03031-f003:**
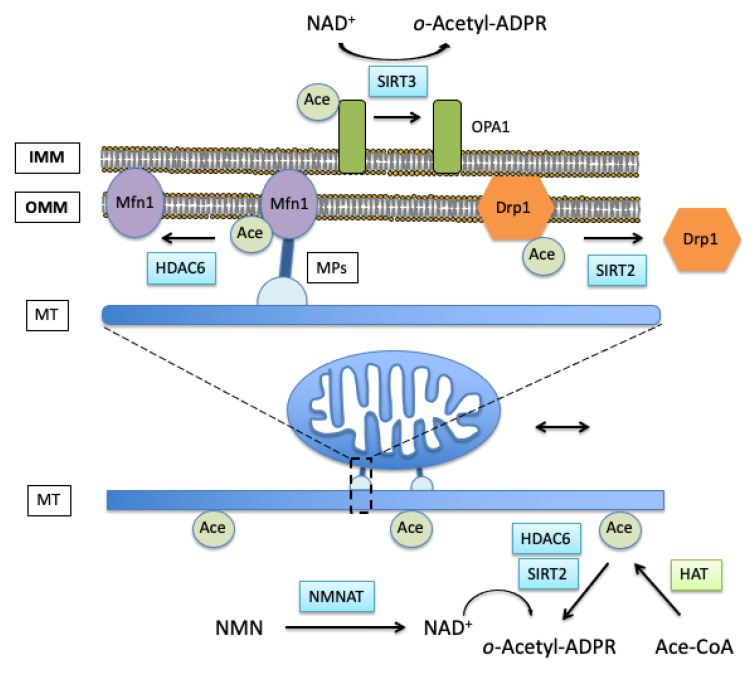
NAD^+^-dependent effect on acetylation-induced modulation of mitochondrial dynamics. Nicotinamide mononucleotide (NMN) serves as precursor for NAD^+^ syntheses by nicotinamide mononucleotide adenylyl transferase (NMNAT). NAD^+^ is a substrate for class III deacetylases (sirtuins) that transfer the acetyl group from acetylated proteins to the ADP-ribose (ADPR) moiety of NAD^+^ forming *o*-acetyl-ADP-ribose (o-Acetyl-ADPR). Increased cytosolic and mitochondrial NAD^+^ levels increase the activity of cytosolic SIRT2, and mitochondrial SIRT3, thus, modulating the acetylation of microtubules (MT) and proteins controlling the mitochondrial fission and fusion processes (Mfn1, Drp1, OPA1). Microtubules and Mfn1 can be also deacetylated by HDAC6. Acetylated Mfn1 can interact with motor proteins (MPs), enabling mitochondrial movement along the microtubules.

**Table 1 cells-10-03031-t001:** NAD^+^ -dependent deacetylases Sirtuins.

Sirtuin	Localization	Activity	Reference
SIRT1	Cytosol, nucleus	deacetylase	[88]
SIRT2	Cytosol	deacetylase	[89]
SIRT3	Mitochondria	deacetylase	[90]
SIRT4	Mitochondria	ADP-ribosyl transferase	[85]
SIRT5	Mitochondria	desuccinylase	[91]
SIRT6	nucleus	ADP-ribosyl transferase	[92]
SIRT7	nucleus	deacetylase	[93]

**Table 2 cells-10-03031-t002:** Fission and fusion proteins modulated by acetylation.

Dynamics Protein	Acetylated Lysine	Acetylation	Reference
Drp1	K642	NAD^+^-dependent	[90]
Fis1	ND	TSA and Nam ^1^	[95]
Mfn1	ND	SIRT1/SIRT2	[93]
Mfn2	ND	SIRT1/SIRT2	[164]
OPA1	ND	SIRT3	[92]
α-tubulin	K40	αTAT1/SIRT2/HDAC6	[134,136]
actin	K50, K61, K328	HDAC6	[139]

^1^ Acetylation is affected by deacetylases inhibitor trichostatin A (TSA) and nicotinamide (Nam); ND, not determined.

## Data Availability

Not applicable.

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
