# Peer review of "Acetylation in Mitochondria Dynamics and Neurodegeneration"

_cells, 2021, doi:10.3390/cells10113031_

Round 1

Reviewer 1 Report

The dynamic behaviour of the mitochondrial network in cells is of crucial importance for metabolic adaptation, stress responses and developmental processes, and therefore critical for cellular and organismal health. The precise coordination of mitochondrial fusion, fission, and intracellular migration is regulated to a great extent by manifold covalent modifications of the involved proteins. Acetylation has been shown to play a major role here and is the major topic of this manuscript. The authors try to give an overview of the molecular machinery required for mitochondrial dynamics, and the proposed roles of acetylation / deacetylation systems in their control. Finally, they link mitochondrial stress to reduced cellular NAD levels and an altered acetylation status of mitochondrial proteins and discuss how such imbalances may contribute to the pathophysiology of acute brain injury.

The topic of this review is timely and interesting; however, in my opinion this manuscript does not meet the usual quality demands. Major problems are:

  1. Redundancy and lack of organization.

The authors tend to repeat statements twice or even three times in different parts of the manuscript. Sometimes they jump forth and back between topics, like in chapter 3.1., from fission to fusion and back. There seems to be some awareness of this problem, because many sentences start with "As mentioned above..."

  1. Scientific accuracy.

Many explanations and statements in this text are so generalized and superficial that they are simply not correct. Elsewhere, very important aspects are simply left out. I will give a number of examples below.

  1. References.

The choice of references in this manuscript is very confusing. Some citations are indeed seminal contributions to the respective fields, either experimental or conceptual, but they are usually very old. Exciting recent developments and newer fine-cut models and findings are ignored. On the other hand, many references are rather marginal contributions or do not fit to the statement to which they are linked in the text. At the same time many important, ground-breaking papers of the last years are not cited at all. In some cases references are even in the completely wrong place, like e.g. reference 91 in line 191.

Here are some concrete examples:

- Line 32 ff.: Parts of this paragraph are more or less taken from reference 7 with minor modifications. The statement that "mitochondria coordinate (...) stress to the ER" is strange to me.

- Line 58ff.: Why is the striking role of ER-mitochondria contact sites and inverted-formin 2 almost completely ignored?

- Line 68ff.: Fragmented mitochondria do not directly fuse with lysosomes.

- Line 85f.: Why should mitochondrial fragmentation reduce the concentration of enzymes and metabolites in the matrix?

- Line 113ff.: The paragraph on OPA1 regulation is confusing. The authors evoke the impression that small forms of OPA1 are only formed under pathological or stress conditions. This is not true.

- Line 162ff.: The paragraph on the sirtuins in confusing. The information should be put into a table.

- Line 183ff.: This sentence does not make sense to me.

- Line 231: What do the authors mean by "increased Drp1 activity". The enzymatic activity is GTPase.

- Line 239: There are so many beautiful studies and reviews out there on the mitochondrial redox homeostasis and ROS protection systems. Reference 103 alone (!) is a particularly weird choice here.

- Line 353ff.: I am missing a discussion about the question, if mitochondrial fragmentation is a true pathological driver and just a consequence of other pathological processes. Although administration of NMN may modulate the acetylation status of mitochondrial proteins, this may well be a side effect and secondary to redox imbalances due to reduced NAD(P).

- Line 384: NAD does not modulate SIRT3 activity. It is a co-substrate of the enzyme, as pointed out elsewhere in the manuscript.

Author Response

The dynamic behavior of the mitochondrial network in cells is of crucial importance for metabolic adaptation, stress responses and developmental processes, and therefore critical for cellular and organismal health. The precise coordination of mitochondrial fusion, fission, and intracellular migration is regulated to a great extent by manifold covalent modifications of the involved proteins. Acetylation has been shown to play a major role here and is the major topic of this manuscript. The authors try to give an overview of the molecular machinery required for mitochondrial dynamics, and the proposed roles of acetylation / deacetylation systems in their control. Finally, they link mitochondrial stress to reduced cellular NAD levels and an altered acetylation status of mitochondrial proteins and discuss how such imbalances may contribute to the pathophysiology of acute brain injury.

The topic of this review is timely and interesting; however, in my opinion this manuscript does not meet the usual quality demands. Major problems are:

  1. Redundancy and lack of organization.

The authors tend to repeat statements twice or even three times in different parts of the manuscript. Sometimes they jump forth and back between topics, like in chapter 3.1., from fission to fusion and back. There seems to be some awareness of this problem, because many sentences start with "As mentioned above..."

Response:We re-arranged the section 3.1. as suggested.

  1. Scientific accuracy.

Many explanations and statements in this text are so generalized and superficial that they are simply not correct. Elsewhere, very important aspects are simply left out. I will give a number of examples below.

  1. References.

The choice of references in this manuscript is very confusing. Some citations are indeed seminal contributions to the respective fields, either experimental or conceptual, but they are usually very old. Exciting recent developments and newer fine-cut models and findings are ignored. On the other hand, many references are rather marginal contributions or do not fit to the statement to which they are linked in the text. At the same time many important, ground-breaking papers of the last years are not cited at all. In some cases references are even in the completely wrong place, like e.g. reference 91 in line 191.

Response:We apologize for the typo. The correct reference should have been 90 (in the original submission).

Here are some concrete examples:

- Line 32 ff.: Parts of this paragraph are more or less taken from reference 7 with minor modifications. The statement that "mitochondria coordinate (...) stress to the ER" is strange to me.

Response:We have rewritten this section.

- Line 58ff.: Why is the striking role of ER-mitochondria contact sites and inverted-formin 2 almost completely ignored?

Response:We added a section describing the role of inverted formin 2 and actin acetylation in mitochondrial fission process.

- Line 68ff.: Fragmented mitochondria do not directly fuse with lysosomes.

Response:The sentence was corrected. Lysosomes were replaced with autophagosomes.

- Line 85f.: Why should mitochondrial fragmentation reduce the concentration of enzymes and metabolites in the matrix?

Response:We did not state that the concentration of enzymes and metabolites in the matrix is reduced, however, their amount is significantly limited in the heavily fragmented organelles. We modified the wording in this sentence.

- Line 113ff.: The paragraph on OPA1 regulation is confusing. The authors evoke the impression that small forms of OPA1 are only formed under pathological or stress conditions. This is not true.

Response:We clarified this paragraph by stating that both isoforms are present in the mitochondrial matrix under physiologic conditions.

- Line 162ff.: The paragraph on the sirtuins in confusing. The information should be put into a table.

Response:We included a table listing sirtuins with their localization and function.

- Line 183ff.: This sentence does not make sense to me.

Response:This sentence was rewritten and divided into two statements.

- Line 231: What do the authors mean by "increased Drp1 activity". The enzymatic activity is GTPase.

Response:We clarified by stating “…increased Drp1-induced fission”.

- Line 239: There are so many beautiful studies and reviews out there on the mitochondrial redox homeostasis and ROS protection systems. Reference 103 alone (!) is a particularly weird choice here.

Response:We added more references.

- Line 353ff.: I am missing a discussion about the question, if mitochondrial fragmentation is a true pathological driver and just a consequence of other pathological processes. Although administration of NMN may modulate the acetylation status of mitochondrial proteins, this may well be a side effect and secondary to redox imbalances due to reduced NAD(P).

Response:NMN does not change the redox state of NAD(P).

- Line 384: NAD does not modulate SIRT3 activity. It is a co-substrate of the enzyme, as pointed out elsewhere in the manuscript.

Response:Enzyme activity depends on substrate concentration; hence it is modulated by substrate levels.

Reviewer 2 Report

This is an interesting review. It could benefit from an additional summary table showing all mitochondria-related proteins regulated by acetylation, showing the positions of relevant lysines (if known), and the involved acetylases a deacetylases (with references).

Many abbreviation require explanation e.g. line 170 - HATs, Fig. 3 - MPs, line 385 - MTs, line 399 - PTM etc. what impairs readability. I would suggest to avoid such a large number of abbreviations in the text and to give a more detailed explanation in the legends to the figures.

Author Response

This is an interesting review. It could benefit from an additional summary table showing all mitochondria-related proteins regulated by acetylation, showing the positions of relevant lysines (if known), and the involved acetylases a deacetylases (with references).

Response:We included a table listing the fission/fusion-related proteins that are regulated by acetylation.

Many abbreviations require explanation e.g. line 170 - HATs, Fig. 3 - MPs, line 385 - MTs, line 399 - PTM etc. what impairs readability. I would suggest avoiding such a large number of abbreviations in the text and to give a more detailed explanation in the legends to the figures.

Response:Line 170 – HATs; Histone acetyl transferase (HAT) is defined on line 157. We replaced the HATs abbreviation with histone acetyl transferases.

Abbreviation MP is explained in the last sentence of the Fig 3 legend.

Line 385 – MTs; The MTs abbreviation was replaced with microtubules.

Line 399 – PTM; We replaced the PTM with post-translational modifications.

We also reduced the number of abbreviations in the text and figure legends.

Reviewer 3 Report

The review manuscript elaborated by Waddell and colleagues is devoted to present an overview on the mitochondrial dynamics process and how acetylation modulates the activity of fusion/fission machinery. The authors also present an overview on possible treatments strategies based on acetylation inhibition in conditions of brain injury. Overall, this is a timely and interesting review that certainly reaches a wide audience. The review is well-written and the figures nicely complement the text.

Author Response

The review manuscript elaborated by Waddell and colleagues is devoted to present an overview on the mitochondrial dynamics process and how acetylation modulates the activity of fusion/fission machinery. The authors also present an overview on possible treatments strategies based on acetylation inhibition in conditions of brain injury. Overall, this is a timely and interesting review that certainly reaches a wide audience. The review is well-written and the figures nicely complement the text.

Response: We thank you the reviewer for kind words and for acknowledging the significance of the submitted review.

Round 2

Reviewer 1 Report

The revision has clearly improved the manuscript. Thank you.

Author Response

References formatting was corrected.